# Cartographing gravity-mediated scattering amplitudes: scalars and photons

Benjamin Knorr[1]* (ID), Samuel Pirlo[2] (ID), Chris Ripken[3] (ID), Frank Saueressig[2] (ID)

**1** Perimeter Institute for Theoretical Physics,
31 Caroline Street North, Waterloo, ON N2L 2Y5, Canada
**2** Institute for Mathematics, Astrophysics and Particle Physics (IMAPP),
Radboud University Nijmegen, Heyendaalseweg 135, 6525 AJ Nijmegen, The Netherlands
**3** Institute of Physics (THEP),
University of Mainz, Staudingerweg 7, 55128 Mainz, Germany
* bknorr@perimeterinstitute.ca

May 5, 2022

## Abstract

The effective action includes all quantum corrections arising in a given quantum field theory. Thus it serves as a powerful generating functional from which quantum-corrected scattering amplitudes can be constructed via tree-level computations. In this work we use this framework for studying gravity-mediated two-to-two scattering processes involving scalars and photons as external particles. We construct a minimal basis of interaction monomials capturing all contributions to these processes. This classification goes beyond the expansions used in effective field theory since it retains the most general momentum dependence in the propagators and couplings. In this way, we derive the most general scattering amplitudes compatible with a relativistic quantum field theory. Comparing to tree-level scattering in general relativity, we identify the differential cross sections which are generated by the non-trivial momentum dependence of the interaction vertices.

# 1   Introduction

Reconciling gravity with the principles of quantum mechanics is one of the major challenges in theoretical physics to date. It is then an intriguing question whether this unification can be achieved within the framework of quantum field theory, underlying our theoretical understanding of particle physics, or requires the introduction of new physics concepts. The gravitational asymptotic safety programme [1–3] (including Dynamical Triangulations [4] and Causal Dynamical Triangulations [5,6]), non-local ghost-free gravity [7], and Hořava-Lifshitz gravity [8–10] clearly advocate the first viewpoint. Generically, one may then wonder about the phenomenological implications resulting from a given fundamental starting point. A systematic understanding of this connection is of key importance when aiming towards corroborating (or falsifying) a given quantum gravity programme.

A pivotal element in connecting the fundamental formulation to its phenomenology is the effective action $\Gamma$. By definition, the propagators and vertices contained in $\Gamma$ include all quantum corrections. Perturbatively, $\Gamma$ can be understood as a power series in $\hbar$ with the lowest order corrections provided by the Tr-log-formula for the one-loop effective action [2,11]. The problem of determining phenomenological consequences can then be broken into two steps, a) computing $\Gamma$ from first principles and b) extracting predictions from $\Gamma$. If the latter can be derived based on the most general form of $\Gamma$, one can use this for a rather straightforward comparison of predictions made by distinguished quantum gravity programmes, at least for the ones based on the principles of quantum field theory.

This perspective motivates a detailed study of the effective action itself. Since the computation of $\Gamma$ from first principles is a notoriously hard problem, tantamount to solving the theory, it is useful to understand which parts of $\Gamma$ actually enter a given observable and what is the most general form this observable can take based on the prerequisite that it has been derived from an effective action. The gravitational form factor programme initiated in [12] and subsequently extended in [13–15]

strives for a systematic investigation of this question for quantum field theories containing gravity and matter degrees of freedom. In particular, [13] studied the gravity-mediated scattering of scalar particles which led to an interesting proposal for realising the asymptotic safety mechanism at the level of gauge-invariant amplitudes [16].

The present work contributes to this programme by extending the construction of the most general gravity-mediated scattering amplitudes for external scalar fields [13, 16] by including an Abelian gauge field (a.k.a. the photon). Since the amplitudes are directly related to experimentally observable cross sections, they are independent of unphysical choices, such as the parameters used in the gauge-fixing. Concretely, we construct the most general (gravity-mediated) two-to-two scattering amplitude with external scalar and gauge fields in a flat Lorentzian background spacetime. In the case of scalar-graviton scattering, this scattering amplitude serves as a model for light bending around a heavy mass, while the pure-photon process models light-by-light scattering [17–19]. As a starting point, we identify all terms in the effective action, including their general momentum dependence, that contribute to these processes. In comparison to the scalar construction, this classification is significantly more involved since the indices carried by the Abelian field strength tensor as well as the accompanying Bianchi identities imply rather intricate equivalence relations between action monomials. Taking these symmetries into account in a systematic way, we then arrive at a minimal basis of interaction terms. All contractions of the field monomials are supplemented by a form factor which encodes the most general momentum dependence of the corresponding interaction. Given that the basis is distilled from a vast number of interaction monomials, our choice is not unique but constitutes a convenient starting point for the construction of amplitudes.

On this basis, we derive the most general two-to-two scattering amplitudes obtainable from the effective action, explicitly tracking the polarisations of the external photons. We observe that, in comparison to tree-level scattering analysed within classical general relativity in a flat Minkowski-background, certain amplitudes become non-trivial owed to the inclusion of the momentum-dependent form factors. These could provide interesting observational channels where the presence of form factors could actually be observed on experimental grounds.

The rest of the work is organised as follows. The minimal basis of interaction terms in $\Gamma$ contributing to the two-to-two scattering processes is constructed in section 2. The most general scattering amplitude resulting from this setting is determined in subsection 3.1 and we discuss some of its properties in subsection 3.2. We close with a brief discussion and outlook in section 4. Technical details on the implementation of our classification algorithm have been relegated to Appendix A, while our notation and conventions on polarisation and momentum vectors are collected in Appendix B.

## 2 The effective action for two-to-two scattering

In this section, we will present the most general effective action contributing to the two-to-two scattering of scalar fields and photons mediated by gravitons. This section is structured as follows. We will begin with an overview of the field content, symmetries and conventions in subsection 2.1. The resulting effective action is given in subsection 2.2. We conclude with discussing possible extensions of our result in subsection 2.3. Technical details regarding the classification of form factors are relegated to Appendix A.

## 2.1 Overview

We will begin with a brief introduction to the effective action formalism and form factors. By definition, the effective action $\Gamma$ encodes the full quantum dynamics of a quantum field theory (QFT). Observables such as scattering cross sections are computed from $\Gamma$ by considering tree-level Feynman diagrams. Both classical interactions and loop corrections are then encoded in form factors. In flat spacetime, these manifest themselves as momentum-dependent functions entering vertices and propagators. Using the Fourier transform, these can be translated to position space, yielding operator-valued functions of partial derivatives. This is straightforwardly generalised to curved spacetime by replacing partial derivatives by covariant ones.

Since $\Gamma$ encodes the resummed quantum corrections to propagators and vertices, it is agnostic about the underlying bare action. Hence, parameterising the most general $\Gamma$ compatible with field content and symmetries allows to describe a broad class of QFTs in a systematic way.

We obtain $n$-point functions from $\Gamma$ by taking $n$ functional derivatives with respect to the fields. In the case of gravity, we implement the functional derivative with respect to the metric $g_{\mu\nu}$ by expanding around the Minkowski metric $\eta_{\mu\nu}$. The graviton $h_{\mu\nu}$ is then defined by

$$g_{\mu\nu} = \eta_{\mu\nu} + h_{\mu\nu}. \tag{1}$$

Inverting the two-point function then gives the effective propagators of the theory, while $n$-vertices are identical to the $n$-point function. Since the $n$-point function is obtained by taking $n$ functional derivatives, it is generally determined by terms in $\Gamma$ containing $n$ fields. In the following, we will denote by $\Gamma_{\Phi^n\Psi^m}$ the building block of $\Gamma$ containing $n$ fields of type $\Phi$ and $m$ fields of type $\Psi$, contributing to a mixed vertex with $n$ $\Phi$-legs and $m$ $\Psi$-legs.

Vertices containing gravitons form a notable exception to this classification. Due to the appearance of $\sqrt{-g}$, any term in $\Gamma$ will be nonlinearly coupled to gravity, and therefore contributes to every vertex containing graviton legs. However, since we expand around a Minkowski background, only terms that contain at most $n$ curvature tensors will contribute to a vertex with $n$ gravitons. With this in mind, we adopt the convention that $\Gamma_{h^2}$ contains up to two curvature tensors, while for $m \geq 1$, $\Gamma_{h^n\Psi^m}$ contains exactly $n$ curvature tensors and $m$ fields of type $\Psi$.

In order to parameterise the most general scattering event, the coupling constant associated to any building block is promoted to a momentum-dependent function, called a form factor. In general, a form factor is a function of the independent contractions of all covariant derivatives in position space. Integration by parts allows to reduce the number of arguments of the form factor. For a form factor acting on $\Phi_1 \cdots \Phi_n$, we denote the arguments of the form factor as

$$\int \Phi_1 f(\Delta)\Phi_2, \qquad\qquad \int f(\Delta_1, \Delta_2, \Delta_3)\Phi_1\Phi_2\Phi_3, \tag{2}$$

and

$$\int f\left(\{-D_i \cdot D_j\}_{1 \leq i < j \leq n}\right)\Phi_1 \cdots \Phi_n, \quad n \geq 4. \tag{3}$$

Here we denoted by $\Delta = -g^{\mu\nu}D_\mu D_\nu$ the covariant d'Alembertian of the covariant derivative $D$ associated to the Levi-Civita connection of $g_{\mu\nu}$. The subscript on each operator denotes the field that it acts on, i.e., $D_1(\Phi_1\Phi_2) = (D_1\Phi_1)\Phi_2$, etc. Throughout this paper, we will often suppress the arguments of form factors to ease the notation.

In the following, we will consider the effective action for a scalar field $\phi$, an Abelian gauge field $A$, hereafter simply called photon, and the graviton $h$. The effective action is constrained by

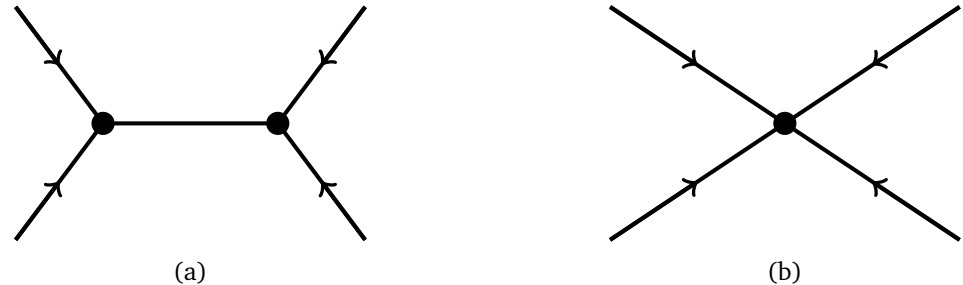

(a) (b)

Figure 1: Tree-level Feynman diagrams contributing to a two-to-two scattering process. In both diagrams, each external line can depict either a scalar or a photon. The effective vertices are denoted by a black circle. We adopt the convention that the momentum of external particles point into the diagram, as shown by the arrows. (a): particle-mediated interaction. The intermediate virtual particle can be either a scalar, photon or graviton. Depending on the labelling of the external legs, this includes $s, t$ and $u$ channels. (b): four-vertex interaction.

the symmetries of the theory. We will assume that $\phi$ satisfies a $\mathbb{Z}_2$-symmetry and is uncharged, while the photon is subject to a U(1) gauge symmetry. In addition, the full action is invariant under diffeomorphisms.

The symmetries strongly reduce the possible tensor structures that can appear in $\Gamma$. Since all particles are uncharged, but live on a curved spacetime, diffeomorphism symmetry dictates that all spacetime curvature quantities are built from the metric and the covariant derivative $D_\mu$. Furthermore, the only U(1)-invariant object is given by the field strength tensor

$$F_{\mu\nu} = \partial_\mu A_\nu - \partial_\nu A_\mu \,. \tag{4}$$

Hence, all structures containing photons are built from this tensor. The field strength tensor satisfies a Bianchi identity

$$D_{[\alpha}F_{\mu\nu]} = 0 \,, \tag{5}$$

where the brackets denote complete antisymmetrisation over the indices. Finally, from $\mathbb{Z}_2$-symmetry we deduce that all terms in the action must contain an even number of $\phi$-fields.

Our aim is to compute the amplitudes of two-to-two particle scattering processes where the external particles are given by scalars or photons. The amplitudes receive contributions from two types of diagrams, depicted in Figure 1. In the first type, the interaction is mediated by a virtual particle which can be a scalar, photon or graviton. The interaction in the second type is given by a four-vertex. Thus, the building blocks for the diagram can be obtained from the two-, three- and four-point functions, and it suffices to parameterise $\Gamma$ up to fourth order in the fields. The effective action $\Gamma$ can then be written as

$$\Gamma[h, \phi, A] \simeq \Gamma_{h^2} + \Gamma_{\phi^2} + \Gamma_{A^2} + \Gamma_{hA^2} + \Gamma_{h\phi^2} + \Gamma_{A\phi^2} + \Gamma_{A^3} + \Gamma_{\phi^4} + \Gamma_{A^4} + \Gamma_{A^2\phi^2} + \Gamma_{\mathrm{gf},h} + \Gamma_{\mathrm{gf},A} \,, \tag{6}$$

where $\simeq$ denotes that the right-hand side is complete up to terms containing more than four fields. The explicit expressions for the building blocks can be found in equations (7)-(18).

In order to obtain well-defined propagators, the action in (6) includes gauge fixing actions $\Gamma_{\mathrm{gf},h}$ and $\Gamma_{\mathrm{gf},A}$ for the graviton and photon, respectively. For the graviton, we employ a de Donder-type

gauge fixing, given by

$$\Gamma_{\text{gf},h} = \frac{1}{32\pi G_N \alpha} \int d^4x \left( \partial^\mu h_{\mu\nu} - \frac{1+\beta}{4} \partial_\nu h^\sigma{}_\sigma \right) \left( \partial_\rho h^{\rho\nu} - \frac{1+\beta}{4} \partial^\nu h^\tau{}_\tau \right), \tag{7}$$

while we gauge-fix the photon by a Lorenz-type gauge fixing,

$$\Gamma_{\text{gf},A} = \frac{1}{2\xi} \int d^4x \left( \partial_\mu A^\mu \right) \left( \partial_\nu A^\nu \right). \tag{8}$$

Here $G_N$ denotes Newton's constant. We will leave the parameters $\alpha$, $\beta$ and $\xi$ general to keep track of gauge (in)dependence of the scattering amplitudes.

## 2.2 Classification of the effective action

Our task is now to parameterise the most general form of each building block. We will first present the part of the action contributing to propagators, before moving to the building blocks contributing to three-vertices and four-vertices. Clearly, the number of admissible terms in $\Gamma$ grows rapidly with increasing complexity of the index structure. Furthermore, special care must be taken not to overcount action terms, since some of them can be related by partial integration or by special relations such as Bianchi identities. To account for this, we have employed a classification algorithm to generate all possible tensor structures, and reduce the number of terms in the action to a minimal set. Details with regard to this algorithm are given in Appendix A.

### 2.2.1 Building blocks up to two fields

We start with building blocks that contribute to propagators only. For the scalar and photon, these are given by

$$\Gamma_{\phi^2} = \frac{1}{2} \int d^4x \sqrt{-g} \, \phi f_{\phi\phi}(\Delta) \phi, \tag{9}$$

$$\Gamma_{A^2} = \frac{1}{4} \int d^4x \sqrt{-g} \, F_{\mu\nu} f_{FF}(\Delta) F^{\mu\nu}. \tag{10}$$

These terms generalise the standard kinetic terms of the scalar and photon fields. The graviton two-point function is obtained by expanding the action to second order in $h$ around the Minkowski metric. Hence, only terms containing at most two curvature tensors will contribute. This gives the action

$$\Gamma_{h^2} = \frac{1}{16\pi G_N} \int d^4x \sqrt{-g} \left[ -R - \frac{1}{6} R f_{RR}(\Delta) R + \frac{1}{2} C_{\mu\nu\rho\sigma} f_{CC}(\Delta) C^{\mu\nu\rho\sigma} \right]. \tag{11}$$

Here $C_{\mu\nu\rho\sigma}$ denotes the Weyl tensor, and we have set the cosmological constant to zero. This ensures that the Minkowski metric is an on-shell solution to the vacuum equation of motion.

### 2.2.2 Building blocks up to three fields

We continue with building blocks including up to three fields. These building blocks contribute to the three-point vertex entering the virtual particle mediated diagrams.

**Gravity-matter vertices**   In this sector, we have vertices containing scalars and vectors. The graviton-scalar vertex is generated by

$$\Gamma_{h\phi^2} = \int \mathrm{d}^4 x \sqrt{-g} \left[ f_{R\phi\phi} R\phi\phi + f_{\mathrm{Ric}\phi\phi} R^{\mu\nu}(D_\mu\phi)(D_\nu\phi) \right], \tag{12}$$

while the graviton-photon vertex is obtained from

$$\begin{aligned}
\Gamma_{hA^2} = \int \mathrm{d}^4 x \sqrt{-g} \Big[ & f_{RFF} R F_{\alpha\beta} F^{\alpha\beta} + f_{\mathrm{Ric}FF} R^{\alpha\beta} F_\alpha{}^\gamma F_{\beta\gamma} + f_{\mathrm{Rm}FF} R_{\alpha\beta\gamma\delta} F^{\alpha\beta} F^{\gamma\delta} \\
& + f_{D^2RFF} (D^\alpha D^\beta R) F_\alpha{}^\gamma F_{\beta\gamma} + f_{D^2\mathrm{Ric}FF} (D^\alpha D^\beta R^{\gamma\delta}) F_{\alpha\gamma} F_{\beta\delta} \\
& + f_{\mathrm{Ric}D^2FF} R^{\gamma\delta} (D^\alpha D^\beta F_{\alpha\gamma}) F_{\beta\delta} + f_{\mathrm{Ric}DFDF} R^{\gamma\delta} (D^\alpha F_{\alpha\gamma})(D^\beta F_{\beta\delta}) \Big].
\end{aligned} \tag{13}$$

Here we have suppressed the derivative-dependence according to the conventions described in Appendix A.

**Photon-scalar vertex**   We now consider the terms in the action that contribute to the $(A\phi\phi)$ vertex. We have a single form factor that contributes:

$$\Gamma_{A\phi^2} = \int \mathrm{d}^4 x \sqrt{-g} f_{F\phi^2} F^{\alpha\beta}(D_\alpha\phi)(D_\beta\phi). \tag{14}$$

Note that a non-vanishing contribution requires that $f_{F\phi^2}(\Delta_1, \Delta_2, \Delta_3)$ is anti-symmetric in its last two arguments.

**Three-photon vertex**   Finally, we have a three-photon vertex:

$$\Gamma_{A^3} = \int \mathrm{d}^4 x \sqrt{-g} \left[ f_{F^3} F_\alpha{}^\beta F_\beta{}^\gamma F_\gamma{}^\alpha + f_{FDFDF} F^{\gamma\delta}(D^\alpha F_{\alpha\gamma})(D^\beta F_{\beta\delta}) \right]. \tag{15}$$

Again $f_{FDFDF}(\Delta_1, \Delta_2, \Delta_3)$ must be anti-symmetric in its last two arguments. This completes our description of the building blocks including up to three fields.

### 2.2.3   Building blocks up to four fields

We conclude this subsection with the building blocks contributing to the four-point vertices. Since we are not considering scattering with external gravitons, these building blocks involve four matter fields only. The four-scalar vertex is generated by a single tensor structure,

$$\Gamma_{\phi^4} = \int \mathrm{d}^4 x \sqrt{-g} f_{\phi^4} \phi\phi\phi\phi. \tag{16}$$

The four-photon vertex is obtained from

$$\begin{aligned}
\Gamma_{A^4} = \int \mathrm{d}^4 x \sqrt{-g} \Big[ & f_{F^2F^2} F_{\alpha\beta} F^{\alpha\beta} F_{\gamma\delta} F^{\gamma\delta} + f_{F^4} F_\alpha{}^\beta F_\beta{}^\gamma F_\gamma{}^\delta F_\delta{}^\alpha \\
& + f_{FFDFDF_1} F_\alpha{}^\gamma F^{\delta\zeta}(D^\alpha F_{\beta\delta})(D^\beta F_{\gamma\zeta}) + f_{FFDFDF_2} F_\beta{}^\gamma F^{\delta\zeta}(D^\alpha F_{\alpha\gamma})(D^\beta F_{\delta\zeta}) \\
& + f_{FFDFDF_3} F_{\alpha\beta} F^{\gamma\delta}(D^\alpha F_\gamma{}^\zeta)(D^\beta F_{\delta\zeta}) + f_{FFDFDF_4} F_\beta{}^\gamma F_{\alpha\gamma}(D^\alpha F^{\delta\zeta})(D^\beta F_{\delta\zeta}) \\
& + f_{FFD^2FD^2F} F_{\alpha\gamma} F_{\beta\delta}(D^\alpha D^\beta F^{\zeta\kappa})(D^\gamma D^\delta F_{\zeta\kappa}) \Big].
\end{aligned} \tag{17}$$

Finally, we have a building block that contributes to the two-scalar-two-photon vertex:

$$
\begin{aligned}
\Gamma_{A^2\phi^2} = \int \mathrm{d}^4 x \sqrt{-g} \Big[ & f_{FF\phi^2} F_{\alpha\beta} F^{\alpha\beta} \phi\phi \\
& + f_{FFD\phi D\phi} F_\alpha{}^\gamma F_{\gamma\beta} (D^\alpha\phi)(D^\beta\phi) + f_{FFD^2\phi\phi} F_\alpha{}^\gamma F_{\gamma\beta} (D^\alpha D^\beta\phi)\phi \\
& + f_{FFD^2\phi D^2\phi} F_{\alpha\gamma} F_{\beta\delta} (D^\alpha D^\beta\phi)(D^\gamma D^\delta\phi) \Big].
\end{aligned}
\tag{18}
$$

## 2.3 Discussion

We conclude this section with a brief discussion of the effective action presented above. The building blocks appearing in $\Gamma$ denote the most general action compatible with the field content and symmetries that we imposed from the start. This representation is by no means unique, as one can apply integration by parts and Bianchi identities to each term. However, the total number of form factors is independent of the way the tensor structures are chosen.

The number of form factors is therefore a sensible benchmark to compare $\Gamma$ to existing computations. Here we compare our result to computations of the trace of the nonlocal heat kernel in covariant perturbation theory [20–22]. It is expected that in the trace of the heat kernel, all possible tensor structures compatible with field content and symmetries are generated. Indeed, we find that the number of form factors in building blocks up to three fields matches the result given in [22].

Some remarks are in order here. First, in an expansion around flat spacetime, the form factors contain both classical and quantum contributions [23]. In particular, they capture the information from eikonal scattering, where the scattered particles stay essentially on-shell, and the momentum transfer is negligible. Second, our result in addition captures the form factors for matter up to four fields. Given the extreme complexity of the nonlocal heat kernel, this result is not available in covariant perturbation theory. Third, in [22], any tensor structure containing a d'Alembertian acting on a Riemann tensor was removed using the Bianchi identity (45). In the case of $\Gamma_{h^3}$, this yields inverse d'Alembertians acting on Ricci tensors. The resulting form factors therefore contain non-analyticities, which we exclude from our parameterisation. Instead, we keep track of tensor structures containing Riemann tensors, *cf.* the third term in eq. (13). At the level of the amplitudes, this results in a non-local overcompleteness, which can be seen by the fact that the form factors $f_{D^2\mathrm{Ric}FF}$ and $f_{\mathrm{Rm}FF}$ appear in a fixed ratio everywhere. This choice of a semi-local basis can break down if there are logarithmic contributions to $f_{\mathrm{Rm}FF}$, which we will however not discuss further here. A motivation to choose our basis is the fact that the coupling related to the local monomial $R^{\mu\nu\rho\sigma} F_{\mu\nu} F_{\rho\sigma}$ is essential, while replacing it via the non-local relation might suggest that the corresponding non-local coupling is inessential [24, 25].

We conclude this section with a brief discussion of possible extensions. In our analysis, we have excluded tensor structures containing the dual field strength tensor $\tilde{F}_{\mu\nu} = \frac{1}{2}\epsilon_{\mu\nu\rho\sigma} F^{\rho\sigma}$. In perturbative QFT, such terms arise from fermionic loops. Since we do not consider fermionic matter, it is self-consistent to consider $\Gamma$ without $\tilde{F}_{\mu\nu}$.

## 3 Scattering in Quantum Field Theory

We will now present the scattering amplitudes of two-to-two scattering processes involving photons and scalar particles described by the action constructed in section 2. We start by presenting

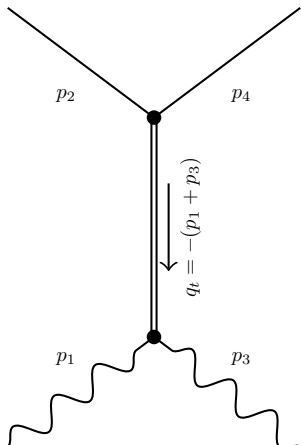

Figure 2: The $t$-channel Feynman diagram that contributes to the $\gamma\phi \to \gamma\phi$ scattering. The external solid lines correspond to scalar legs, external wavy lines correspond to photon legs, the internal double line corresponds to the gauge-fixed graviton propagator obtained from (11). The black dots indicate the three-point $h\phi\phi$- and $h\gamma\gamma$-vertices encoded in (12) and (13), respectively.

the amplitudes for the processes $\gamma\phi \to \gamma\phi$, $\phi\phi \to \gamma\gamma$ and $\gamma\gamma \to \gamma\gamma$ with generic form factors. Subsequently, we construct the cross sections associated to these amplitudes and comment on the low-energy behaviour. The pure scalar process $\phi\phi \to \phi\phi$ has already been discussed in [13].

## 3.1 Scattering amplitudes involving external photons

### 3.1.1 $\gamma\phi \to \gamma\phi$ scattering

We first consider the process $\gamma\phi \to \gamma\phi$. The diagrams that contribute are the $t$-channel and the four-point vertex. In the case of the $t$-channel diagram, the exchanged particle can either be a scalar, photon or a graviton. However, the vertex derived from the action (14) with the two scalar legs on-shell equals zero. Similarly, this vertex evaluated with one of the scalars and the photon on-shell also vanishes. Hence, the $s$-channel diagram with a virtual scalar also does not contribute to this process. As a consequence, the process is governed by the $t$-channel contribution with a virtual graviton, depicted in Figure 2. The full amplitude for the process $\gamma\phi \to \gamma\phi$ is then given by the combination

$$\mathcal{A} = \mathcal{A}_t + \mathcal{A}_4. \tag{19}$$

Denoting right- and left-handed photons using $+$'s and $-$'s respectively, the corresponding independent helicity amplitudes can then be computed using the relations in Appendix B. We will use the convention that all momenta are ingoing, and helicities are adjusted accordingly.[1] The

---

[1]The helicities of outgoing particles with respect to outgoing momenta are thus obtained by flipping the sign.

calculation was performed with the help of the Mathematica package suite *xAct* [26, 27], yielding

$$
\mathcal{A}_t^{++} = \quad -\frac{2\pi G_N}{3}\left(s^2 - 4su + u^2 + 2m_\phi^4\right)t\,\frac{\left(1 - tf_{\mathrm{Ric}D\phi D\phi}\right)\left(tf_{D^2\mathrm{Ric}FF} - 4f_{\mathrm{Rm}FF} - f'_{FF}(0)\right)}{t\left(1 + tf_{CC}(t)\right)} \tag{20}
$$

$$
+\frac{2\pi G_N}{3}t^2\,\frac{\left(t + 2m_\phi^2 + 2t\left(t - m_\phi^2\right)f_{\mathrm{Ric}D\phi D\phi} - 12tf_{R\phi\phi}\right)}{t\left(1 + tf_{RR}(t)\right)}
$$

$$
\times\left(6tf_{D^2RFF} + tf_{D^2\mathrm{Ric}FF} - 24f_{RFF} - 6f_{\mathrm{Ric}FF} - 4f_{\mathrm{Rm}FF} + 2f'_{FF}(0)\right),
$$

$$
\mathcal{A}_t^{+-} = 4\pi G_N\left(su - m_\phi^4\right)\frac{\left(1 - tf_{\mathrm{Ric}D\phi D\phi}\right)\left(2 + t^2 f_{D^2\mathrm{Ric}FF} - 2tf_{\mathrm{Ric}FF} - 4t\,f_{\mathrm{Rm}FF}\right)}{t\left(1 + tf_{CC}(t)\right)}\,. \tag{21}
$$

Here we suppressed the on-shell arguments of the form factors to lighten the notation in the following way:

$$
\begin{aligned}
f_{R\phi\phi} &= f_{R\phi\phi}(t, m_\phi^2, m_\phi^2), & f_{\mathrm{Ric}D\phi D\phi} &= f_{\mathrm{Ric}D\phi D\phi}(t, m_\phi^2, m_\phi^2), \\
f_{RFF} &= f_{RFF}(t, 0, 0), & f_{\mathrm{Ric}FF} &= f_{\mathrm{Ric}FF}(t, 0, 0), \\
f_{D^2RFF} &= f_{D^2RFF}(t, 0, 0), & f_{D^2\mathrm{Ric}FF} &= f_{D^2\mathrm{Ric}FF}(t, 0, 0), \\
f_{\mathrm{Rm}FF} &= f_{\mathrm{Rm}FF}(0, 0). &
\end{aligned} \tag{22}
$$

Furthermore, $\mathcal{A}$ receives a contribution from the four-point vertex. This reads

$$
\mathcal{A}_4^{++} = \frac{1}{4}t\Big[8f_{FF\phi\phi}(t, s, u) - 2sf_{FFD\phi D\phi}(t, s, u) \tag{23}
$$

$$
+ 2m_\phi^2 f_{FFD^2\phi\phi}(t, s, u) - (su - m_\phi^4)f_{FFD^2\phi D^2\phi}(t, s, u)\Big] + (s \leftrightarrow u);
$$

$$
\mathcal{A}_4^{+-} = \frac{1}{4}(su - m_\phi^4)\Big[2f_{FFD\phi D\phi}(t, s, u) - 2f_{FFD^2\phi\phi}(t, s, u) - tf_{FFD^2\phi D^2\phi}(t, s, u)\Big] + (s \leftrightarrow u). \tag{24}
$$

Here, we suppressed half of the arguments of the form factors following the rule

$$
f_{PQRS}(a, b, c) = f_{PQRS}\left(\frac{a}{2}, \frac{b - m_\phi^2}{2}, \frac{c - m_\phi^2}{2}, \frac{c - m_\phi^2}{2}, \frac{b - m_\phi^2}{2}, \frac{a}{2} - m_\phi^2\right). \tag{25}
$$

The two remaining helicity configurations can then be obtained by parity transformations. This gives

$$
\mathcal{A}^{++} = \mathcal{A}^{--}, \qquad \mathcal{A}^{+-} = \mathcal{A}^{-+}. \tag{26}
$$

### 3.1.2 $\phi\phi \to \gamma\gamma$ scattering

As a convenient by-product of the $\gamma\phi \to \gamma\phi$ amplitude, we obtain the amplitude of the $\phi\phi \to \gamma\gamma$ process by crossing symmetry. Since the polarisation channels do not change, the amplitudes are obtained from (19) by interchanging $s \leftrightarrow t$. To avoid doubling lengthy formulas, we will refrain from repeating the explicit expressions here.

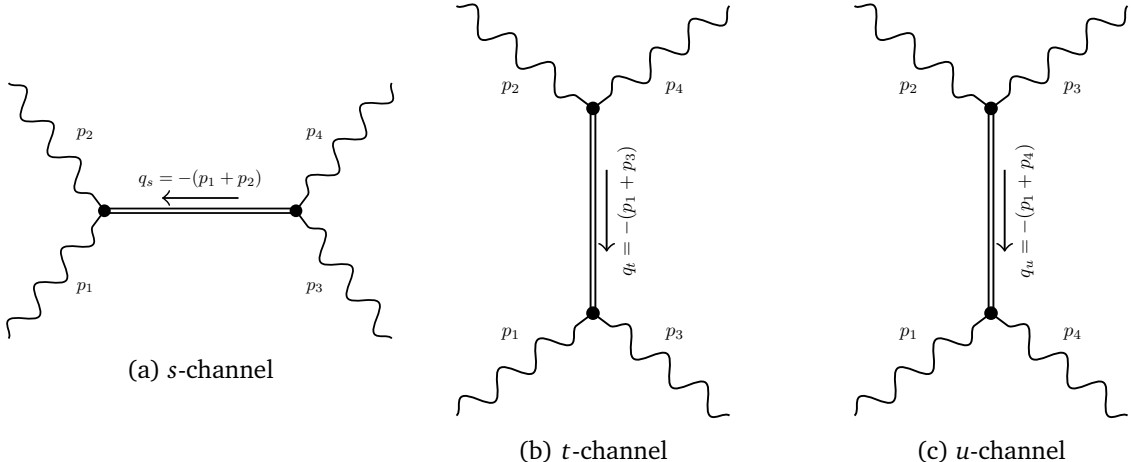

Figure 3: Feynman diagrams encoding the graviton-mediated contribution to the $\gamma\gamma \to \gamma\gamma$ scattering amplitude. The wavy lines correspond to photon legs, the internal double lines corresponds to the gauge-fixed graviton propagator obtained from (11). The black dots indicate the three-point $h\gamma\gamma$-vertex encoded in (13).

### 3.1.3 $\gamma\gamma \to \gamma\gamma$ scattering

We will now consider four-photon scattering. Again, we have particle-mediated and four-point contributions to the scattering amplitude. For the particle-mediated diagram, the exchanged particle is either a photon or a graviton. Computing the vertices arising from the action (15), and setting the two external photon legs on-shell shows that these vertices vanish. Therefore, the only contribution comes from a graviton-exchanged diagram.

The graviton-mediated contribution is given by the combination of $s$-, $t$- and $u$-channel diagrams, depicted in Figure 3. Therefore, the full amplitude for this process is given by

$$\mathcal{A} = \mathcal{A}_s + \mathcal{A}_t + \mathcal{A}_u + \mathcal{A}_4. \tag{27}$$

In computing the right-hand side of the above expression, it is sufficient to evaluate only the $t$-channel contribution $\mathcal{A}_t^{\gamma\gamma}$. The $s$- and $u$-channel diagrams are obtained from the $t$-channel diagram by applying crossing symmetry, interchanging $t \leftrightarrow s$ and $t \leftrightarrow u$, respectively. The amplitudes can then be organised by their helicity configurations. We have the following classes:

$$
\begin{aligned}
\text{I:} \quad & \mathcal{A}^{+--+} = \mathcal{A}^{-++-}, \\
\text{II:} \quad & \mathcal{A}^{++++} = \mathcal{A}^{----}, \\
\text{III:} \quad & \mathcal{A}^{++--} = \mathcal{A}^{--++}, \\
\text{IV:} \quad & \mathcal{A}^{+-+-} = \mathcal{A}^{-+-+}, \\
\text{V:} \quad & \mathcal{A}^{+++-} = \mathcal{A}^{++-+} = \mathcal{A}^{+-++} = \mathcal{A}^{-+++} = \mathcal{A}^{---+} = \mathcal{A}^{--+-} = \mathcal{A}^{-+--} = \mathcal{A}^{+---}.
\end{aligned}
\tag{28}
$$

The $t$-channel contributions to these expressions read

$$\text{I:} \quad \mathcal{A}_t = 2\pi G_N s^2 \frac{\left(2 + t^2 f_{D^2\text{Ric}FF} - 2t\, f_{\text{Ric}FF} - 4t\, f_{\text{Rm}FF}\right)^2}{t\left(1 + t\, f_{CC}(t)\right)}, \tag{29}$$

|  | I | II | III | IV | V |
|---|---|---|---|---|---|
|  | $+--+$ | $++++$ | $++--$ | $+-+-$ | $+++-$ |
| $f_{F^2F^2}$ | $4s^2[f(s,t,u)+f(s,u,t)]$ | $4t^2[f(t,s,u)+f(t,u,s)]$ | $4u^2[f(u,s,t)+f(u,t,s)]$ | I+II+III $-2\{su[f(s,t,u)+f(u,t,s)]]$ | 0 |
| $f_{F^4}$ | $s^2$[perm.] | $t^2$[perm.] | $u^2$[perm.] | $+st[f(s,u,t)+f(t,u,s)]$ $+tu[f(t,s,u)+f(u,s,t)]\}$ | 0 |
| $f_{FFDFDF_1}$ | $\frac{1}{2}s^2\{t[f(s,t,u)-f(t,s,u)]$ $+u[f(s,u,t)-f(u,s,t)]\}$ | $\frac{1}{2}t^2\{s[f(t,s,u)-f(s,t,u)]$ $+u[f(t,u,s)-f(u,t,s)]\}$ | $\frac{1}{2}u^2\{s[f(u,s,t)-f(s,u,t)]$ $+t[f(u,t,s)-f(t,u,s)]\}$ | $\frac{1}{2}stu$[perm.] | 0 |
| $f_{FFDFDF_2}$ | 0 | 0 | 0 | 0 | 0 |
| $f_{FFDFDF_3}$ | 0 | 0 | 0 | $stu$[perm.] | $\frac{1}{4}stu$[perm.] |
| $f_{FFDFDF_4}$ | $s^2[tf(s,u,t)+uf(s,t,u)]$ | $t^2[sf(t,s,u)+uf(t,u,s)]$ | $u^2[sf(u,t,s)+tf(u,s,t)]$ | I+II+III | $-\frac{1}{2}stu$[perm.] |
| $f_{FFD^2FD^2F}$ | $-\frac{1}{2}s^2tu[f(s,t,u)+f(s,u,t)]$ | $-\frac{1}{2}t^2su[f(t,s,u)+f(t,u,s)]$ | $-\frac{1}{2}u^2st[f(u,s,t)+f(u,t,s)]$ | I+II+III | $\frac{1}{2}\cdot$IV |

Table 1: Different contributions of the four-photon vertex to $\mathcal{A}_4$. The function $f$ should be read as $f(a,b,c)=f_I\left(\frac{a}{2},\frac{b}{2},\frac{c}{2},\frac{c}{2},\frac{b}{2},\frac{a}{2}\right)$. By [perm.], we denote the sum of $f(s,t,u)$ and the five permutations of its arguments. The 16 polarisation configurations can be obtained by interchanging ($+\leftrightarrow-$) following the scheme in (28). The total amplitude for each polarisation configuration is obtained by summing the contribution from each form factor in the respective column.

$$
\text{II}=\text{IV}:\quad \mathcal{A}_t=\frac{\pi}{3}G_N t^2(s^2-4su+u^2)\frac{\left(-t\,f_{D^2\mathrm{Ric}FF}+4f_{\mathrm{Rm}FF}+f'_{FF}(0)\right)^2}{t\left(1+t\,f_{CC}(t)\right)} \tag{30}
$$

$$
-\frac{\pi}{3}G_N t^4\frac{\left(6t\,f_{D^2RFF}+t\,f_{D^2\mathrm{Ric}FF}-24f_{RFF}-6f_{\mathrm{Ric}FF}-4f_{\mathrm{Rm}FF}+2f'_{FF}(0)\right)^2}{t\left(1+t\,f_{RR}(t)\right)},
$$

$$
\text{III}:\quad \mathcal{A}_t=2\pi G_N u^2\frac{\left(2+t^2 f_{D^2\mathrm{Ric}FF}-2t\,f_{\mathrm{Ric}FF}-4t\,f_{\mathrm{Rm}FF}\right)^2}{t\left(1+t\,f_{CC}(t)\right)}, \tag{31}
$$

$$
\text{V}:\quad \mathcal{A}_t=2\pi G_N su\left(-t^2 f_{D^2\mathrm{Ric}FF}+4t\,f_{\mathrm{Rm}FF}+t\,f'_{FF}(0)\right) \tag{32}
$$

$$
\times\frac{\left(2+t^2 f_{D^2\mathrm{Ric}FF}-2t\,f_{\mathrm{Ric}FF}-4t\,f_{\mathrm{Rm}FF}\right)}{t\left(1+t\,f_{CC}(t)\right)}.
$$

Finally, we present the four-point diagram. In this case, amplitudes for the different helicity configurations are all distinct. The explicit expressions of the four-point diagrams are rather lengthy, but are summarised in Table 1. The expression for $\mathcal{A}_4$ is obtained by adding the entries in a given column. This completes the discussion of all amplitudes.

## 3.2 Cross sections

At this point, we can draw some interesting conclusions from the computed effective scattering amplitudes. To this end, it is instructive to convert the scattering amplitudes into full-fledged observables. We will consider the differential scattering cross section. For a two-to-two scattering process, the differential cross section of the polarisation configuration $a$ is straightforwardly computed in the centre-of-mass frame by

$$
\left(\frac{\mathrm{d}\sigma^a}{\mathrm{d}\Omega}\right)_{\mathrm{CM}}=\frac{1}{64\pi^2 s}\left|\mathcal{A}^a\right|^2. \tag{33}
$$

In this frame, the differential cross section can be expressed in the centre-of-mass momentum $p=|\mathbf{p}|$ and the scattering angle $\theta$. The total cross section is then obtained by integrating over the

scattering angle $\theta$ and the azimuthal angle $\varphi$.

In order to determine which form factors can be accessed most easily by scattering experiments, it is instructive to expand the cross section around small three-momentum $p$.[2] Then for the process $\gamma\phi \to \gamma\phi$, we find

$$\frac{d\sigma_{\gamma\phi}^{++}}{d\Omega} = \frac{1}{9}G_N^2 \left(4f_{\mathrm{Rm}FF}(0,0) + f'_{FF}(0)\right)^2 \left(m_\phi^6 - 2m_\phi^5 p\right) + \mathcal{O}(p^2), \tag{34}$$

$$\frac{d\sigma_{\gamma\phi}^{+-}}{d\Omega} = G_N^2 \left(m_\phi^2 + 2m_\phi p\right) \frac{(1-\cos\theta)^2}{(1+\cos\theta)^2} + \mathcal{O}(p^2). \tag{35}$$

Interestingly, to leading order the expanded cross section receives no contributions from the four-point diagram. The $(+-)$ configuration gives the general relativity (GR) result, and it corresponds to the helicity-conserving process. On the other hand, the leading order contribution to the cross section of the helicity-flipping process, corresponding to the $(++)$ configuration, is non-zero only in the presence of form factors.

For the process $\gamma\gamma \to \gamma\gamma$, the cross sections expand to:

$$\frac{d\sigma_{\gamma\gamma}^{\mathrm{I}}}{d\Omega} = 16G_N^2 p^2 \frac{1}{(1+\cos\theta)^2} + \mathcal{O}(p^4), \tag{36}$$

$$\frac{d\sigma_{\gamma\gamma}^{\mathrm{II}}}{d\Omega} = \frac{1}{4\pi^2} \left(4f_{F^2F^2}(\mathbf{0}) + 3f_{F^4}(\mathbf{0})\right)^2 (1+\cos\theta)^4 p^6 + \mathcal{O}(p^8), \tag{37}$$

$$\frac{d\sigma_{\gamma\gamma}^{\mathrm{III}}}{d\Omega} = G_N^2 p^2 \frac{(1-\cos\theta)^4}{(1+\cos\theta)^2} + \mathcal{O}(p^4), \tag{38}$$

$$\frac{d\sigma_{\gamma\gamma}^{\mathrm{IV}}}{d\Omega} = \frac{1}{\pi^2} \left(4f_{F^2F^2}(\mathbf{0}) + f_{F^4}(\mathbf{0})\right)^2 (3+\cos^2\theta)^2 p^6 + \mathcal{O}(p^8), \tag{39}$$

$$\frac{d\sigma_{\gamma\gamma}^{\mathrm{V}}}{d\Omega} = 4G_N^2 \left(4f_{\mathrm{Rm}FF}(0,0) + f'_{FF}(0)\right)^2 (1-\cos\theta)^2 p^6 + \mathcal{O}(p^8). \tag{40}$$

Here we have denoted $\mathbf{0} = (0,0,0,0,0,0)$ for brevity. We see that for four-photon scattering, the leading order contributions in the I and III channels are given by the GR result. The V channel receives a contribution at leading order from the graviton-mediated diagram. The II and IV channels receive contributions from the four-point diagrams. Their differential cross sections can be distinguished both from a different dependence on the scattering angle and a different coupling constant. Here the leading orders are induced by form factors. At low three-momentum, the cross section is dominated by the channels I and III, with the form factor processes being suppressed by an additional factor of $p^4$.

At this point, several remarks are in order. First, the scattering cross sections describe the scattering of scalars and photons in a flat Minkowski background. Owed to the presence of the particles, which give rise to a non-trivial energy momentum tensor, this background is not on-shell. We then observe that the cross sections exhibit clear deviations from the scattering based

---

[2]This expansion requires the assumption that the form factors admit an analytic expansion around $p = 0$. For quantum corrections related to massless particles, such as the photon, we expect that the form factors include logarithmic contributions. Typically, these are resolved by resummation techniques. Inspecting the cross sections below, we note that a logarithmic contribution in the form factor $f_{FF}$ could induce additional terms at a lower order in $p$. In particular, $d\sigma_{\gamma\gamma}^{\mathrm{V}}/d\Omega$ would then start at the same order in $p$ as the channels I and III. We also observe that $f_{FF}$ appears in a fixed combination with the $f_{\mathrm{Rm}FF}$ form factor in both cross sections $d\sigma_{\gamma\phi}/d\Omega$ and $d\sigma_{\gamma\gamma}/d\Omega$, which suggests that these functions are related in the infrared. This is an incentive for future investigations.

on GR in the same background. In the latter case, all form factors are zero and the expressions (34), (37) and (40) vanish. The form factor corrections then contain both classical corrections (appearing without $\hbar$) encoding classical curved GR, *e.g.*, through the eikonal approximation, as well as genuine quantum corrections (coming with powers of $\hbar$). Within the effective action, both contributions are contained in the form factors.

Second, it is interesting to ask whether the cross sections may be measured experimentally. In this context, we note that Standard Model of particle physics processes also induce non-trivial contributions to the matter form factors considered here. A well-known example is the four-photon interaction of the Euler-Heisenberg Lagrangian [28]. These contributions will typically overshadow any quantum gravity contribution due to the suppression of the latter by powers of the Planck mass. However, this would not be the case if the form factors include non-local terms associated with inverse powers of the momentum. We can thus constrain the functional form of the form factors by comparing the measurement of the differential cross sections to the prediction of the Standard Model, at least in principle.

Third, the total cross section of the scattering process is infinite, due to the physical divergence in the forward scattering limit, $\theta = \pi$. This divergence already appears in the minimally coupled case and is caused by the presence of massless gravitons. From the low-energy expansion, we see that this cannot be ameliorated by contributions from the form factors, unless these contain non-analyticities. It is expected that these divergences are cured when the contribution of soft gravitons emitted by the external particles is taken into account [29].

# 4  An outlook on gravitational observables

In this work, we have parameterised the most general amplitude for a two-to-two scattering process involving scalars and Abelian gauge fields mediated by gravitons. As a physical application, these amplitudes describe the bending of light around a heavy mass. The momentum-dependence of vertices and propagators is encoded in form factors associated to monomials in the effective action. We have presented an algorithm to compute a minimal basis of these monomials. Although the index structure of Abelian gauge fields admits a large number of possible tensor structures, the number of independent monomials turns out to be rather small.

We conclude this paper with an outlook on gravitational observables that employ the form factor formalism. The case of gravity-mediated scalar scattering has already been discussed in [13]. The present setup can be generalised in the following directions. First of all, the classification of effective action monomials can be expanded to include (fermionic) matter fields. A classification of the fermionic effective action up to zeroth order in the curvature can be found in [12]. Of special interest here are form factors that couple curvature to the chiral components of the fermions. This allows to parameterise any chiral symmetry breaking induced by gravity.

The classification can also be applied in the bosonic sector, in particular for graviton scattering. For a two-to-two graviton scattering process, it is then necessary to parameterise all action monomials up to cubic order in the curvature for graviton-mediated scattering, and to quartic order to map out the graviton four-point function. Although the appearance of a field containing two spacetime indices will greatly increase the complexity, we expect no conceptual difficulties in generalising the procedure presented in this paper.

A second application of the form factor formalism lies in the actual computation of gravitational form factors. This can be done either perturbatively or non-perturbatively. In a perturba-

tive setting, the gravitational corrections to the propagator form factors have been computed [30] and give rise to a logarithmic momentum-dependence at first order in loop corrections. To our knowledge, the form factors contributing to the gravity-matter vertices have not been computed completely, see [31–33] for partial results. However, the one-loop scattering amplitude does give information about the analytic structure of these form factors; the appearance of square roots of the momentum in the one-loop scattering amplitude can be attributed to square roots in the gravity-matter form factors, parameterising classical GR effects [23]. By applying suitable approximations to the form factors, our results thus directly connect to gravitational effective field theory. Consistency conditions imposed on the EFT like unitarity and causality allow to constrain the Wilson coefficients derived from such an expansion [34].

Going beyond perturbation theory, form factors can be computed from first principles in the realm of asymptotically safe gravity. Here we distinguish two procedures. First, the form factors can be computed in the continuum. This method uses functional renormalisation group techniques [35] to capture the non-perturbative quantum corrections to form factors. This scheme has been used to compute the momentum dependence of the graviton propagator [36–42], and in later works to investigate the momentum dependence of the three and four point functions [43,44], see [45] for a review. These computations have proven to be very complex, but we expect significant progress using essential renormalisation group techniques [24, 25]. Second, one can use data from lattice approaches to quantum gravity as an input for form factors. In [46], data from Causal Dynamical Triangulations has been used to partially fit the gravity form factor $f_{RR}$.

Finally, physics implications from a more phenomenological perspective have been discussed in the context of the tanh model [16]. This approach implements Asymptotic Safety along the original idea of Weinberg [47,48] and provides a proof of principle that Lorentzian Asymptotic Safety can be realised at the level of the effective action. The latter then connects computations tracking the momentum dependence using the functional renormalisation group to gauge-invariant observables.

We close the discussion of applications of the form factor formalism to gravitational observables with the following cautious remark. In principle, it is tempting to use the amplitudes to reconstruct the spacetime geometry, since to lowest order the Newtonian potential derived from an amplitude agrees with the potential in the geodesic equation [31]. Beyond the leading order, this identification is known to fail [15,49]. Thus, in general the reconstruction of the full geometry from an amplitude is significantly more involved.

# Acknowledgements

BK and CR acknowledge the hospitality at Radboud University during the final stages of this project. We thank Renata Ferrero, Carlo Pagani and Martin Reuter for useful discussions.

**Funding information**   BK acknowledges support by Perimeter Institute for Theoretical Physics. Research at Perimeter Institute is supported in part by the Government of Canada through the Department of Innovation, Science and Economic Development and by the Province of Ontario through the Ministry of Colleges and Universities.

# A Technical details regarding the classification of the effective action

In this appendix, we will provide a detailed overview of the algorithm that we used to classify the effective action $\Gamma$ given in section 2. While writing down the most general effective action is straightforward for scalar fields only, deciding upon a minimal set of monomials in $\Gamma$ becomes complicated if one considers more fields or fields with internal tensor structure. In this work, we provide a systematic way for choosing such a set.

This classification algorithm consists of two steps. First, one constructs the set of all possible action monomials that may contribute to a given scattering process, and is compatible with field content and symmetries of the theory. Secondly, we reduce this set to a basis. Typically, the set constructed in the first step is overcomplete, due to operations such as partial integration or the application of identities such as the Bianchi identity. In the following subsections, we describe each step in more detail.

## A.1 Generating action monomials

We will begin with generating a complete set of monomials. Having decided upon a scattering process, we determine the $n$-point functions that contribute to this process. This gives the building blocks $\Gamma_{\Phi^n \Psi^m}$ that need to be parameterised. Here $\Phi$ and $\Psi$ stand for any of the fields taking part in the chosen process. If the desired $n$-point function contains scalar legs, this fixes the number of scalar fields that have to be present in each term of $\Gamma_{\Phi^n \Psi^m}$. Since photon and graviton fields are subject to U(1) gauge symmetry and diffeomorphism symmetry, photon fields can only appear in the form of a field strength tensor $F_{\mu\nu}$, while gravitational interactions are generated by the metric volume element $\sqrt{-g}$, covariant derivatives, and (contractions of) the Riemann tensor $R_{\mu\nu\rho\sigma}$. Thus, the number of photon legs present in the $n$-point function determines the number of field strength tensors. Since we expand around a flat background, the number of graviton legs gives an upper bound for the number of curvature tensors.

Having determined how many fields appear in each building block, and in which form, we can construct all possible action monomials. This is done as follows. Suppose that we want all action monomials containing $n_R$ Riemann tensors, $n_F$ field strength tensors, and $n_\phi$ scalar fields. Then we go through the following steps:

1) Write down a string of the Riemann and field strength tensors and scalar fields, each with uncontracted indices. In total, this object has $4n_R + 2n_F$ free indices.

2) Write down all possible ways of distributing a covariant derivatives over the string of fields, each with a unique index. Thus, we obtain tensors containing from zero up to $4n_R + 2n_F$ covariant derivatives.

3) Contract the free indices in all possible combinations.

4) Equip each tensor structure with a unique form factor, given by an operator-valued function of contracted covariant derivatives, and integrate over spacetime.

It is convenient to use tensor algebra software to perform these steps. Especially in step 3), a large number of tensor structures in generated. We have exploited *xAct*'s `AllContractions` method [26, 27, 50–52] to generate these structures.

## A.2 Reduction to a minimal set

Having generated the set of all action monomials that are compatible with field content and symmetries, we reduce this set to a minimal basis. Let $\{\Phi_i\}_{i=1\ldots n}$ denote a set of tensors that is linear in one matter field or spacetime curvature. We note that some action monomials can be related via the following transformations:

- Contraction of derivatives. Contracted derivatives can be absorbed into the form factor.

- Permutation of labels. Monomials that are related by permutations of $\Phi_i$ are equivalent. Hence,

$$\int f(i,j)\Phi_1 \cdots \Phi_i \cdots \Phi_j \cdots \Phi_n = \int f(j,i)\Phi_1 \cdots \Phi_j \cdots \Phi_i \cdots \Phi_n, \tag{41}$$

where we have abbreviated the $i$ and $j$ dependence of the form factor by $f(i,j)$.

- Integration by parts. A derivative acting on $\Phi_i$ can be integrated by parts:

$$\int f\, D_i(\Phi_1 \cdots \Phi_i \cdots \Phi_n) \simeq -\int f \sum_{i\neq j} D_j(\Phi_1 \cdots \Phi_i \cdots \Phi_n). \tag{42}$$

Here we dropped curvature terms that arise from commuting the covariant derivative with the form factor.

- Bianchi identities. The Riemann tensor satisfies

$$R_{[\alpha\beta\gamma]}{}^{\delta} = 0 \qquad \text{and} \qquad D_{[\mu}R_{\alpha\beta]\gamma}{}^{\delta} = 0, \tag{43}$$

while the field strength tensor satisfies

$$D_{[\alpha}F_{\beta\gamma]} = 0. \tag{44}$$

These identities allows to permute indices within Riemann and field strength tensors.

We successively apply the transformations listed above. This leads to a significant reduction of the number of independent terms. Schematically, this is done as follows. We can represent the action monomials and transformations between them as a graph with nodes and edges, respectively. We can find a minimal set of monomials by choosing a representative from each connected component of this graph. In this work, we have used the following guidelines to choose a representative:

- The representative contains the smallest number of derivatives outside the form factor.

- The representative contains the curvature tensor with the smallest number of indices. That is, we eliminate monomials containing a Riemann tensor in favor of a Ricci tensor, and a Ricci tensor in favor of a Ricci scalar.

- At least one tensor $\Phi_i$ is free of derivatives outside the form factor.

- Derivatives outside the form factor are distributed as symmetrically as possible.

Remarkably, the remaining set of independent action monomials is rather small compared to the total number of possible tensor structures, and is presented in subsection 2.2.

At this stage, several remarks are in order. First, the second Bianchi identity allows to express any tensor structure containing a d'Alembertian acting on a Riemann tensor in terms of Ricci tensors. To be precise, we have

$$D^2 R_{\rho\sigma\mu\nu} \simeq 2D_\rho D_{[\mu} R_{\nu]\sigma} - 2D_\sigma D_{[\mu} R_{\nu]\rho} \,, \tag{45}$$

where $\simeq$ denotes equality up to terms quadratic in the curvature. This means that any form factor acting on a tensor structure containing a Riemann tensor cannot sustain any d'Alembertian acting on this Riemann tensor. In this work, this only applies to the form factor $f_{\mathrm{Rm}FF}$ associated with the action monomial

$$\int \sqrt{-g} f_{\mathrm{Rm}FF}(\Delta_2, \Delta_3) R_{\alpha\beta\gamma\delta} F^{\alpha\beta} F^{\gamma\delta} \,. \tag{46}$$

Here the form factor does not depend on a d'Alembertian acting on $R_{\alpha\beta\gamma\delta}$. Furthermore, in $\Gamma_{h^2}$ we have chosen the form factor $f_{CC}$ associated to a (Weyl)$^2$ tensor structure, in favor of a (Ric)$^2$ tensor structure. This ensures that the form factor $f_{CC}$ appears in the spin-2 part of the graviton propagator only.

Second, we comment about keeping track of the form factors in applying the transformations. Since partial integration and the Bianchi identity only affect the tensor structure, it is tempting to forget about the form factor and only consider how contractions of the field tensors are related. However, permutation of labels *does* affect the form factor, which a priori does not have any symmetries. Therefore, it is important to keep track of changes in the form factor as well. A minimal working example is given by the the monomial

$$\int f \, F_\alpha{}^\beta F^{\alpha\gamma} (D_\beta D_\gamma \phi) \phi \,, \tag{47}$$

which is manifestly asymmetric in the two scalar fields. One might consider to replace this by the more symmetric tensor structure

$$\int f(1,2,3,4) (D_\alpha F^{\alpha\gamma}) (D_\beta F^\beta{}_\gamma) \phi \phi \,. \tag{48}$$

However, using partial integration and the Bianchi identity repeatedly, we can show that this is equal to

$$\int f(1,2,3,4) (D_\alpha F^{\alpha\gamma}) (D_\beta F^\beta{}_\gamma) \phi \phi = \int \frac{1}{2} f(1,2,3,4) \left( D_1 \cdot D_3 + D_1 \cdot D_4 - D_2^2 \right) F_{\alpha\beta} F^{\alpha\beta} \phi \phi$$

$$+ \int \left( f(1,2,3,4) + f(1,2,4,3) \right) \left( F_\alpha{}^\gamma F_{\beta\gamma} (D^\alpha D^\beta \phi) \phi + F_\alpha{}^\gamma F_{\beta\gamma} (D^\alpha \phi)(D^\beta \phi) \right) . \tag{49}$$

Thus, it follows that the tensor structure (48) can only be mapped to (47) with a *symmetric* form factor. Using (48) as basis element instead of (47) will therefore result in an incomplete basis.

In line with this example, we note that any permutation symmetry of the tensor structure imposes the same permutation symmetry on the form factor. In order to simplify our notation, we do not keep track of this symmetrisation explicitly.

# B Conventions

In this appendix we collect our conventions. Generally, we work with a metric with mostly minus signature. In particular, the Minkowski metric reads

$$
\eta_{\mu\nu} = \begin{pmatrix} 1 & 0 & 0 & 0 \\ 0 & -1 & 0 & 0 \\ 0 & 0 & -1 & 0 \\ 0 & 0 & 0 & -1 \end{pmatrix}.
\tag{50}
$$

At every vertex, all momenta are considered as ingoing. This means that momentum conservation takes the form

$$
\sum_i p_i = 0.
\tag{51}
$$

Our on-shell conditions are

$$
p^2 = 0,
\tag{52}
$$

for photons, and

$$
p^2 = m_\phi^2,
\tag{53}
$$

for scalars. For the two-to-two scattering processes considered in this work we adopt the notation that the ingoing quantities carry labels $1, 2$ whereas the outgoing quantities carry labels $3, 4$. The Mandelstam variables are defined as

$$
s = (p_1 + p_2)^2, \qquad t = (p_1 + p_3)^2, \qquad u = (p_1 + p_4)^2.
\tag{54}
$$

## B.1 Polarisation vectors

To project external photon lines onto physical states, we need to define polarisation vectors. We introduce them by defining a spatial (circular) polarisation vector, and lift it to a four-vector by adding a vanishing time component. We label the helicity by $+$ and $-$, and the corresponding polarisation vectors are complex conjugates of each other. For example, for an ingoing polarisation vector,

$$
\epsilon_\mu^{\text{in}-} = \epsilon_\mu^{\text{in}+*}.
\tag{55}
$$

Since we are working in the centre-of-mass frame, we only need a single ingoing and a single outgoing polarisation vector to completely describe two-to-two scattering processes.

By definition, the inner product of the polarisation vector with the appropriate momentum vector vanishes:

$$
p_1^\mu \epsilon_\mu^{\text{in}\pm} = p_2^\mu \epsilon_\mu^{\text{in}\pm} = p_3^\mu \epsilon_\mu^{\text{out}\pm} = p_4^\mu \epsilon_\mu^{\text{out}\pm} = 0.
\tag{56}
$$

A general polarisation is a linear combination of the circular $\pm$ polarisations. In this way,

$$
\epsilon_\mu^{1,2} = \frac{1 - \lambda_{1,2}}{2} \epsilon_\mu^{\text{in}+*} + \frac{1 + \lambda_{1,2}}{2} \epsilon_\mu^{\text{in}+},
\tag{57}
$$

$$
\epsilon_\mu^{3,4} = \frac{1 - \lambda_{3,4}}{2} \epsilon_\mu^{\text{out}+*} + \frac{1 + \lambda_{3,4}}{2} \epsilon_\mu^{\text{out}+},
\tag{58}
$$

where $\lambda_{1,2,3,4}$ indicates the helicity.

To simplify concrete computations, we introduce a coordinate system to specify the polarisation vectors, and to compute all necessary scalar products. For this, let $\mathbf{p}$ be the ingoing three-momentum, defined to be along the $x$-axis, and $\mathbf{q}$ the outgoing three-momentum, both in the centre-of-mass frame, so that

$$\mathbf{p} = \left(\sqrt{\mathbf{p}^2}, 0, 0\right), \qquad \mathbf{q} = \left(\sqrt{\mathbf{q}^2}\cos\theta, \sqrt{\mathbf{q}^2}\sin\theta, 0\right). \tag{59}$$

Here $\theta$ is the scattering angle. The corresponding unit vectors are related by the rotation matrix

$$\mathbf{R} = \begin{pmatrix} \cos\theta & -\sin\theta & 0 \\ \sin\theta & \cos\theta & 0 \\ 0 & 0 & 1 \end{pmatrix}. \tag{60}$$

With this, we define the ingoing polarisation vector within the $(y-z)$-plane

$$\epsilon_\mu^{\text{in}+} = \frac{1}{\sqrt{2}}(0, 0, 1, -\mathbf{i}). \tag{61}$$

The outgoing polarisation vector is the rotated ingoing polarisation vector,

$$\epsilon_\mu^{\text{out}+} = \frac{1}{\sqrt{2}}(0, -\sin\theta, \cos\theta, -\mathbf{i}). \tag{62}$$

We can now compute the necessary scalar products:

$$\epsilon^{\text{in}\mu+}\epsilon_\mu^{\text{in}+} = \epsilon^{\text{in}\mu+*}\epsilon_\mu^{\text{in}+*} = \epsilon^{\text{out}\mu+}\epsilon_\mu^{\text{out}+} = \epsilon^{\text{out}\mu+*}\epsilon_\mu^{\text{out}+*} = 0, \tag{63}$$

$$\epsilon^{\text{in}\mu+*}\epsilon_\mu^{\text{in}+} = \epsilon^{\text{out}\mu+*}\epsilon_\mu^{\text{out}+} = -1, \tag{64}$$

$$\epsilon^{\text{in}\mu+}\epsilon_\mu^{\text{out}+} = \epsilon^{\text{in}\mu+*}\epsilon_\mu^{\text{out}+*} = \frac{1}{2}(1 - \cos\theta), \tag{65}$$

$$\epsilon^{\text{in}\mu+*}\epsilon_\mu^{\text{out}+} = \epsilon^{\text{in}\mu+}\epsilon_\mu^{\text{out}+*} = -\frac{1}{2}(1 + \cos\theta), \tag{66}$$

$$p_1^\mu \epsilon_\mu^{\text{out}+} = p_1^\mu \epsilon_\mu^{\text{out}+*} = \frac{1}{\sqrt{2}}\sqrt{\mathbf{p}^2}\sin\theta, \tag{67}$$

$$p_2^\mu \epsilon_\mu^{\text{out}+} = p_2^\mu \epsilon_\mu^{\text{out}+*} = -\frac{1}{\sqrt{2}}\sqrt{\mathbf{p}^2}\sin\theta, \tag{68}$$

$$p_3^\mu \epsilon_\mu^{\text{in}+} = p_3^\mu \epsilon_\mu^{\text{in}+*} = -\frac{1}{\sqrt{2}}\sqrt{\mathbf{q}^2}\sin\theta, \tag{69}$$

$$p_4^\mu \epsilon_\mu^{\text{in}+} = p_4^\mu \epsilon_\mu^{\text{in}+*} = \frac{1}{\sqrt{2}}\sqrt{\mathbf{q}^2}\sin\theta. \tag{70}$$

To translate these into an invariant language in terms of Mandelstam variables, we have to distinguish between the different processes.

## B.2  $\gamma\phi \to \gamma\phi$ scattering

For the process of a photon and a scalar scattering to another photon and a scalar, the momenta in the centre-of-mass frame read

$$p_{1\mu} = \left(\sqrt{\mathbf{p}^2}, \mathbf{p}\right), \tag{71}$$

$$p_{2\mu} = \left( \sqrt{\mathbf{p}^2 + m_\phi^2}, -\mathbf{p} \right), \tag{72}$$

$$p_{3\mu} = \left( -\sqrt{\mathbf{p}^2}, \mathbf{q} \right), \tag{73}$$

$$p_{4\mu} = \left( -\sqrt{\mathbf{p}^2 + m_\phi^2}, -\mathbf{q} \right). \tag{74}$$

In this case, $|\mathbf{p}| = |\mathbf{q}|$, and the scattering angle $\theta$ is defined by

$$\mathbf{p} \cdot \mathbf{q} = \mathbf{p}^2 \cos \theta. \tag{75}$$

For the Mandelstam variables, this implies

$$s = m_\phi^2 + 2\mathbf{p}^2 + 2\sqrt{\mathbf{p}^2 \left( \mathbf{p}^2 + m_\phi^2 \right)} \geq m_\phi^2, \tag{76}$$

$$t = -2\mathbf{p}^2 \left( 1 + \cos \theta \right), \tag{77}$$

$$u = m_\phi^2 - 2\left( \sqrt{\mathbf{p}^2 \left( \mathbf{p}^2 + m_\phi^2 \right)} - \mathbf{p}^2 \cos \theta \right). \tag{78}$$

The squared three-momentum and sine and cosine of the scattering angle can be expressed via

$$\mathbf{p}^2 = \frac{\left( s - m_\phi^2 \right)^2}{4s}, \qquad \cos \theta = -\left( 1 + 2\frac{st}{\left( s - m_\phi^2 \right)^2} \right), \qquad \sin \theta = \frac{2}{\left( s - m_\phi^2 \right)^2} \sqrt{st \left( su - m_\phi^4 \right)}. \tag{79}$$

### B.3   $\phi\phi \to \gamma\gamma$ scattering

For the process of two scalars scattering to two photons, the momenta in the centre-of-mass frame read

$$p_{1\mu} = \left( \sqrt{\mathbf{p}^2 + m_\phi^2}, \mathbf{p} \right), \tag{80}$$

$$p_{2\mu} = \left( \sqrt{\mathbf{p}^2 + m_\phi^2}, -\mathbf{p} \right), \tag{81}$$

$$p_{3\mu} = \left( -\sqrt{\mathbf{q}^2}, \mathbf{q} \right) = \left( -\sqrt{\mathbf{p}^2 + m_\phi^2}, \mathbf{q} \right), \tag{82}$$

$$p_{4\mu} = \left( -\sqrt{\mathbf{q}^2}, -\mathbf{q} \right) = \left( -\sqrt{\mathbf{p}^2 + m_\phi^2}, -\mathbf{q} \right). \tag{83}$$

In this case, $|\mathbf{p}| \neq |\mathbf{q}|$, and the scattering angle $\theta$ is defined by

$$\mathbf{p} \cdot \mathbf{q} = \sqrt{\mathbf{p}^2 \left( \mathbf{p}^2 + m_\phi^2 \right)} \cos \theta. \tag{84}$$

For this process, the Mandelstam variables read

$$s = 4\mathbf{q}^2 = 4\left( \mathbf{p}^2 + m_\phi^2 \right) \geq 4m_\phi^2, \tag{85}$$

$$t = -\left( 2\mathbf{p}^2 + m_\phi^2 + 2\sqrt{\mathbf{p}^2 \left( \mathbf{p}^2 + m_\phi^2 \right)} \cos \theta \right), \tag{86}$$

$$u = -\left( 2\mathbf{p}^2 + m_\phi^2 - 2\sqrt{\mathbf{p}^2 \left( \mathbf{p}^2 + m_\phi^2 \right)} \cos \theta \right). \tag{87}$$

As required for this process, they fulfil the relation

$$s + t + u = \sum_{i=1}^{4} p_i^2 = 2m_\phi^2 \,. \tag{88}$$

One way to express the squared three-momenta and the sine and cosine of the scattering angle in terms of the Mandelstam variables is

$$\mathbf{p}^2 = \frac{s - 4m_\phi^2}{4}\,, \qquad \mathbf{q}^2 = \frac{s}{4}\,, \qquad \cos\theta = \frac{u - t}{\sqrt{s\left(s - 4m_\phi^2\right)}}\,, \qquad \sin\theta = 2\sqrt{\frac{tu - m_\phi^4}{s\left(s - 4m_\phi^2\right)}}\,. \tag{89}$$

### B.4 $\gamma\gamma \to \gamma\gamma$ **scattering**

Finally, we consider the process of two photons scattering to two photons. In this case the momenta in the centre-of-mass-frame read

$$p_{1\mu} = \left(\sqrt{\mathbf{p}^2}, \mathbf{p}\right), \tag{90}$$

$$p_{2\mu} = \left(\sqrt{\mathbf{p}^2}, -\mathbf{p}\right), \tag{91}$$

$$p_{3\mu} = \left(-\sqrt{\mathbf{q}^2}, \mathbf{q}\right) = \left(-\sqrt{\mathbf{p}^2}, \mathbf{q}\right), \tag{92}$$

$$p_{4\mu} = \left(-\sqrt{\mathbf{q}^2}, -\mathbf{q}\right) = \left(-\sqrt{\mathbf{p}^2}, -\mathbf{q}\right). \tag{93}$$

and the scattering angle is defined by

$$\mathbf{p} \cdot \mathbf{q} = \mathbf{p}^2 \cos\theta \,. \tag{94}$$

The Mandelstam variables are

$$s = 4\mathbf{p}^2 \geq 0 \,, \tag{95}$$

$$t = -2\mathbf{p}^2(1 + \cos\theta) \,, \tag{96}$$

$$u = -2\mathbf{p}^2(1 - \cos\theta) \,. \tag{97}$$

As required for this process, they fulfil the relation

$$s + t + u = \sum_{i=1}^{4} p_i^2 = 0 \,. \tag{98}$$

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
