# Peer review of "Cartographing gravity-mediated scattering amplitudes: scalars and photons"

_SciPost Physics_

## Round 1 · Referee Report · Anonymous (Referee 1) · 2022-9-7

Strengths

The general idea of enumerating these form factors is good. The paper represents significant work in organizing the form factors.

Weaknesses

It seems that the authors are excluding important non-local effects. See report.

Report

This paper contains a good deal of content which is potentially useful. However, it seems also incomplete because it neglects the nonlocalities which come with quantum calculations. This is briefly discussed, but is just dismissed without any resolution.

The basic problem is that with quantum corrections there can be inverse powers of d’Alambertians which seem not to be captured in this formfactor expansion. An extreme version of this is seen in the Riegert anomaly action R.J. Riegert, Phys. Lett. 134B (1984) 56 , which has powers of 1/Box^2 . But also the Barvinsky Vilkovisky expansion at third order in the curvature has many logs in the numerators and 1/Box factor in the denominator. Another example is the recent nonlocal action of Donoghue Phys. Rev. D105} 105025 (2022) which have various nonlocal factors when expressed in terms of the curvature. These quantum effects seem common, but they also seem to be missed by the work or the present paper.

The issue is mentioned briefly on page 8. But the authors say only that these effects are excluded from their parametrization. While this is true, it is not sufficient. I understand fully their comment about BV removing the structure with the Riemann tensor. But there are other nonlocalities besides these. These effects do occur regularly. If the authors work is to be useful, they need to explain clearly how their parameterization is to be used in the face of these nonlocalities. The immediate worry is that they are useless if there are sufficient nonlocal terms. Maybe this is too extreme a conclusion – I can’t really see through the effects of these – but it certainly needs more comment that a casual dismissal.

I recommend that the paper be returned to the authors for a full discussion of the non-analytic effects which they are excluding. Please address directly Riegert and Barvinsky-Vilkovisky actions and say how they fit into the formfactor scheme that they are using. . If the effects are being ignored, how are their results to be used? Is there a need to generalize this work to include the quantum nonlocalities.?

Requested changes

See report

---

## Round 1 · Referee Report · Anonymous (Referee 2) · 2022-9-25

Strengths

Clear discussion of form factors and their relation to scattering amplitudes.

Weaknesses

No actual calculation.

Report

The standard way to calculate scattering cross sections in particle physics is to directly derive scattering amplitudes from the relevant Feynman diagrams, at a given loop order.
A less direct route is to first compute the effective action, or rather parts thereof, and then to derive from these the scattering amplitudes. The benefit of this longer route is that the effective action can be used also in other contexts.

This paper deals with the second and easier step, namely the translation of terms in the effective action to amplitudes. It identifies and classifies the relevant terms for the scattering of scalars and Maxwell fields, and then converts them to amplitudes. This discussion is well organized, clearly written and pedagogically useful.

However, when compared with other similar papers, this one appears to be lacking in physical content. For example in arXiv 0812.2729 [gr-qc], 1202.4502 [hep-th], or the recent 2001.10196, both effective action and amplitudes are calculated.
Similarly, the present paper would be greatly improved by including at least some particular examples of processes that are worked out from first principles. Perhaps something can be extracted from already-calculated pieces of the effective action, as given in 2002.10839 [hep-th].
An example of such a calculation can be found in 1006.3808 [hep-th].

---

## Editorial Decision

awaiting_resubmission